# Tackling Antibiotic Resistance with Compounds of Natural Origin: A Comprehensive Review

**DOI:** 10.3390/biomedicines8100405

**Published:** 2020-10-11

**Authors:** Francisco Javier Álvarez-Martínez, Enrique Barrajón-Catalán, Vicente Micol

**Affiliations:** 1Institute of Research, Development and Innovation in Health Biotechnology of Elche (IDiBE), Universitas Miguel Hernández (UMH), 03202 Elche, Spain; f.alvarez@umh.es (F.J.Á.-M.); vmicol@umh.es (V.M.); 2CIBER, Fisiopatología de la Obesidad y la Nutrición, CIBERobn, Instituto de Salud Carlos III (CB12/03/30038), 28220 Madrid, Spain

**Keywords:** natural antimicrobial, antimicrobial resistance, polyphenols, future medicine, natural origin, antibacterial compound, phytochemicals

## Abstract

Drug-resistant bacteria pose a serious threat to human health worldwide. Current antibiotics are losing efficacy and new antimicrobial agents are urgently needed. Living organisms are an invaluable source of antimicrobial compounds. The antimicrobial activity of the most representative natural products of animal, bacterial, fungal and plant origin are reviewed in this paper. Their activity against drug-resistant bacteria, their mechanisms of action, the possible development of resistance against them, their role in current medicine and their future perspectives are discussed. Electronic databases such as PubMed, Scopus and ScienceDirect were used to search scientific contributions until September 2020, using relevant keywords. Natural compounds of heterogeneous origins have been shown to possess antimicrobial capabilities, including against antibiotic-resistant bacteria. The most commonly found mechanisms of antimicrobial action are related to protein biosynthesis and alteration of cell walls and membranes. Various natural compounds, especially phytochemicals, have shown synergistic capacity with antibiotics. There is little literature on the development of specific resistance mechanisms against natural antimicrobial compounds. New technologies such as -omics, network pharmacology and informatics have the potential to identify and characterize new natural antimicrobial compounds in the future. This knowledge may be useful for the development of future therapeutic strategies.

## 1. Introduction

Antimicrobial resistance (AMR) and the inexorable advance of superbacteria poses a great threat to human health worldwide. If this problem is not tackled, the antibiotics we have used with great success so far could become substances unable to help us against infections caused by bacteria, going back to a worrying pre-antibiotic era. According to data from the United Kingdom government [1], 10 million deaths could happen annually due to antibiotic resistance by 2050, becoming one of the leading causes of death in the world (Figure 1).

This problem is known to scientists and institutions around the world, which are organizing to establish protocols to address the problem of antibiotic-resistant microbes. Proof of this was the 2012 Chennai Declaration of India, in which international experts and representatives of medical entities met to draw up action plans in the face of the inexorable advance of the superbugs [2]. Similar initiatives have been promoted from private and public institutions worldwide.

Bacteria use their genetic plasticity to resist attack by antibiotics through mutations, acquisition of genetic material, and alteration of the expression of their genome [3]. In this way, bacteria that survive the attack of an antibiotic become the precursors of the next bacterial generations, further aggravating the problem of resistance. Once antibiotic resistance genes are acquired, they can be passed from one bacterium to another through division processes or by horizontal gene transfer [4]. Horizontal gene transfer processes can occur by transformation, transduction or conjugation with other bacteria. These mechanisms can transfer antibiotic resistance to bacteria that have not been subjected to antibiotic selection pressure, creating reservoirs of resistant bacteria in the environment [5]. In addition, the epistasis of the receptor bacteria plays a fundamental role in the process of acquisition of resistance genes, determining whether these bacteria are capable of maintaining, accumulating and propagating the genetic material [6].

Antibiotic resistance is an example of the enormous capacity for natural evolution and adaptation of bacteria to different environments [7,8]. Although this process seems inevitable, humans have accelerated it through various anthropogenic activities [9,10]. The causes behind the increase in the number of antimicrobial-resistant bacteria in recent years include the misuse of antibiotics in humans and animals, inadequate control of infections in hospitals and clinics or poor hygiene and sanitation [9,10,11]. In addition to the causes mentioned, the problem worsens as there is a drought in the discovery of new antibiotics. The increase in resistance rates in bacteria leads to a decrease in the effectiveness of existing antibiotics, making research in this field unattractive to companies that decide to invest in other types of fields with greater chances of success and benefits [12,13]. This concerning trend can be observed in Figure 2.

In view of this scenario, research on alternative or complementary therapies to traditional antibiotics has emerged strongly. Antimicrobial products of natural origin have been positioned as compounds of great scientific interest due to their enormous chemical variety and intrinsic properties that have promoted their study as a possible therapeutic tool in recent years.

## 2. Methodology

Electronic databases such as PubMed, Scopus and ScienceDirect were used to search scientific contributions until September 2020, using relevant keywords. Search terms included “natural antimicrobial”, “antimicrobial resistance”, “polyphenols”, “future medicine”, “natural origin”, “antibacterial compound”, “phytochemical” and their combinations. Literature focusing on the antimicrobial activity of natural origin compounds against bacteria focusing on antibiotic-resistant strains were identified and summarized.

The term “antimicrobial activity” is used throughout this work to refer to the process of killing or inhibiting the growth of microbes. Usually, this activity is expressed as MIC (minimum inhibitory concentration) values for a given agent. The methods to test microbial susceptibility compiled in this work are in accordance with the guidelines of the European Committee on Antimicrobial Susceptibility Testing (EUCAST) and The Clinical and Laboratory Standards Institute (CLSI). Following the EUCAST guidelines for the reproducibility and reliability of antimicrobial assays, broth dilution or microdilution methods should be used to test microbial susceptibility [14].

## 3. Results

### 3.1. Use of Natural Products as Antimicrobials

Natural products (NPs) make up a heterogeneous group of chemical entities that possess diverse biological activities with various uses in fields such as human and veterinary medicine, agriculture and industry. Molecules from the secondary metabolism of animals, vegetables, bacteria and fungi are classified as NPs, which are not crucial for the producer’s survival under laboratory conditions, but which give him a clear advantage over his competitors in his native habitat [15]. Since the discovery of penicillin, more than 23,000 new NPs have been characterized, many of which have proven to be valuable tools in the field of pharmacology, herbicides, insecticides and more [16].

One of the main sources of antimicrobial NPs is plants. Plant organisms make up most of the biosphere on planet Earth, whose biomass accounts for a percentage greater than 80% of the total biomass [17]. Since their appearance, plants have survived, evolved and adapted to all types of ecosystems and adverse conditions. This adaptive process has led them to develop complex and effective defense systems against external aggressions: predators, abiotic stress and, of course, infections. Being sessile organisms that cannot escape their threats, plants have developed a splendid chemical arsenal in the form of secondary metabolites capable of coping with the most dangerous pathogens [18]. Humanity has made use of the medicinal properties of plants for thousands of years. There is evidence that in the year 5000 BC. the Sumerians already used thyme for its beneficial health properties [19]. The Egyptian Ebers Papyrus dating from around 1500 BC already attributed medicinal properties to plants and spices such as aloe vera, castor bean, garlic, hemp, anise or mustard [20,21]. Other texts such as the Atharva Veda, the Rig Veda and the Sushruta Samhita belonging to Indian Ayurveda, also spoke of the pharmacological properties of plant substances such as turmeric or cannabis [22,23]. Current technology allows us to study the bases of this ancestral knowledge and find therapeutic applications adapted to our time, making plants a source of invaluable therapeutic potential.

Bacteria are another of the main sources of antimicrobial NPs with radical importance during the 20th century. Most of the antibiotics used today in the clinic were discovered thanks to the Waksman platform in the 1940s. Waksman and his students dedicated themselves to growing soil microorganisms to detect and isolate antimicrobial substances. Through this method, they discovered very important antibiotics such as neomycin or streptomycin, for which Waksman received the Nobel Prize in 1952 for Physiology or Medicine [24]. Despite these successes, it should be noted that most existing bacteria are not cultivable in the laboratory using traditional methods. We could find an immense amount of opportunities for the isolation of new antibiotic compounds using a method like Waksman’s combined with new technologies not present decades ago. From this idea, the Small World Initiative was born in 2012, a project in which students from all over the world collect soil samples and look for antibiotic-producing microorganisms in them [25].

Many of the NPs with antibiotic activity have been isolated from bacteria, especially from the genus actinomycetes. In the so-called “Golden Age” of the discovery of new antibiotics, which began in the 40s of the twentieth century, natural products were the star. The isolation of streptomycin from *Streptomyces griseus* in 1944 caused a worldwide surge in which numerous research groups struggled to identify new NPs, especially from samples of soil bacteria. The media were very limited, both in technology and in access to soil samples from remote places. However, another great milestone occurred in 1952, when a sample of soil sent from Borneo allowed *Streptomyces orientalis* to grow, from which vancomycin was extracted. Six years later, vancomycin was used in patients with great success. Unfortunately, this prolific period of discovery of valuable compounds ended the appearance and spread of bacteria resistant to these NPs, such as methicillin-resistant *Staphylococcus aureus* (MRSA) or glycopeptide-resistant enterococci (GREs), since the compounds that worked in the past stopped working with the desired efficiency [26], as observed in Figure 2.

In the 1990s, the pharmaceutical industry concentrated its efforts on other more sophisticated methods of identifying antimicrobial compounds, such as high-throughput screening of synthetic chemical libraries against specific therapeutic targets, many of them discovered from the Human Genome Project. Currently, there is a renewed interest in the discovery of new NPs of different sources since it has a much more advanced technology than that available during the “Golden Age”. Advances in genomics, bioinformatics and mass spectrometry, among others, have elucidated that many of the sources of classical NPs were surprisingly under-exploited and have an enormous and unknown potential for the discovery of new NPs to be used for the discovery of present and tomorrow’s antibiotics [15].

Given the existing problems in the field of antibiotics, in recent years alternative and complementary therapies have emerged that make use of different strategies to deal with new generations of resistant bacteria. The growing interest in this area is reflected in the ascending number of publications related to natural antimicrobials available in the PubMed search engine over the past recent years (Figure 3).

As abovementioned, the molecules with antimicrobial function present in nature have been molded by thousands of years of evolution to maintain their efficacy and selectivity, since they are a key piece for the development of the life of any organism exposed to bacteria. Thanks to these processes of continuous physicochemical adaptation driven by selective pressure, it has been demonstrated that antimicrobial compounds of natural origin generally have a greater capacity for cell penetration, being able to use active bacterial transporters and, in addition, passively pass through the cell membrane [27]. These and other properties that will be discussed below, make NPs a tool of great potential value for the development of novel and effective antibiotic therapies against AMR bacteria.

### 3.2. Main Classes of Natural Antimicrobial Products

NPs are extremely diverse in terms of their chemical structures, properties and mechanisms of action. These agents can be classified according to their original source: animal, bacterial, fungal or vegetal.

#### 3.2.1. Animal Origin

Animals have colonized virtually the entire planet Earth. For thousands of years, they have lived closely with different kinds of bacteria and have faced not a few pathogenic microorganisms. Evolution has shaped animal defense systems to deal with these microscopic threats. In recent years, attention has been focused on identifying which molecules confer resistance and allow certain animals to live in hostile environments with high pollution and pathogenic load, as is the case with certain insects such as cockroaches.

Currently, animals, and especially insects, are one of the main sources of antimicrobial proteins or peptides (AMPs). Since the discovery of AMPs in 1974, more than 150 new AMPs have been isolated or identified, the majority being cationic peptides between 20 and 50 residues in length. These molecules mainly have antimicrobial capacity mediated by disruption of the bacterial plasma membrane, most probably by forming pores or ion channels [28]. Some AMPs also have shown antifungal, antiparasitic or antiviral properties [29]. These AMPs can be divided into four subfamilies with different structures and sequences: the α-helical peptides, such as cecropin, which has a broad spectrum of antimicrobial activity against bacteria of both Gram-positive and Gram-negative bacteria; cysteine-rich peptides, such as insect defensins, which are mainly active against Gram-positive bacteria; proline-rich peptides, such as lebocins, which are active against both Gram-positive and Gram-negative bacteria and some fungi; and finally glycine-rich peptides or proteins, such as attacin, which are effective against Gram-negative bacteria and especially against *Escherichia coli*. These AMPs present a promising basis for the development of medical therapies, however, additional work must be developed to make them more powerful and stable [30]. Moreover, the intrinsic antimicrobial capacity of AMPs can be enhanced by a fusion of peptides to create more potent hybrid ones, such as in the case of attacin from *Spodoptera exigua* and a coleoptericin-like protein from *Protaetia brevitarsis seulensis*, which, when fused, exhibited a greater antimicrobial capacity than its two original peptides [31].

The study of antimicrobial molecules existent in cockroaches (*Periplaneta americana*) has revealed that extracts derived from its brain have a great antimicrobial capacity against MRSA and neuropathogenic *E. coli* K1. Although not all the components of the extract could be accurately identified, a great variety of molecules with known biological activity were found, such as isoquinolines, flavanones, sulfonamides and imidazone among others. A hypothesis about the production of this antimicrobial cocktail in the cockroach brain suggests that there could be a constitutive expression of these antimicrobials to protect the animal’s neural system, since it is the central axis of its survival and a key piece to protect when it is lived in an environment of high pollution and exposure to pathogens and even superbugs [32]. Another example of insect producing antimicrobial molecules against resistant bacteria is *Lucilia cuprina* blowfly maggots. The extract obtained from excretions and secretions from maggots showed mild bacterial growth inhibition. However, using subinhibitory concentrations of this extract in combination with the antibiotic ciprofloxacin enhanced its activity, further delaying the appearance of bacteria resistant to it. The properties of this extract, including the presence of defensins and phenylacetaldehyde, make maggot debridement therapy a promising tool in the treatment of MRSA-infected wounds acquired in hospital [33].

One of the most popular insect-related products worldwide is honey. In addition to its nutritional properties and culinary values, it has antimicrobial capacity against Gram-negative bacteria, such as *E. coli* or *Pseudomonas aeruginosa*, and against Gram-positive bacteria, such as *Bacillus subtilis* or *S. aureus*, including MRSA. The key factors of honey’s antimicrobial activity appear to be the presence of H_2_O_2_, bee defensin-1 and methylglyoxal. The diverse molecular composition of the different honey types that depends on the producing species and the raw material used, exerts also different antimicrobial activities and mechanisms [34]. Another substance produced by bees is propolis, a resinous substance produced by honeybees from plant matter, such as buds or sap. This substance has been used since ancient times, up to 3000 years BC in Egypt thanks to its various biological properties. The main components responsible for its activity are flavonoids, terpene derivatives and phenolic acids, although its composition is variable depending on the geographical area where it occurs. Ethanol extract of propolis produced by *Apis mellifera* in Brazil has demonstrated significant antibacterial capacity against *S. aureus*, *E. coli* and *Enterococcus sp*. [35]. Canadian propolis has also been shown to possess antibacterial capacity against *E. coli* and *S. aureus*, being more effective against the latter [36]. Another product with antimicrobial properties derived from honeybees is royal jelly. It is produced from the mandibular salivary and hypopharyngeal glands of bees aged between 5 and 14 days. Its composition is based on a complex mixture of carbohydrates, proteins, lipids, vitamins and minerals that varies with regional conditions, season, bee’s genetics and postharvest storage conditions. Royal jelly shows antimicrobial activity against both Gram-positive and Gram-negative bacteria, including MDR bacteria such as MRSA. The compounds isolated from royal jelly with activity against Gram-positive bacteria are the peptide royalisin [37], the peptide family of jelleines and 10-hydroxy-2-decenoic acid (10-HDA), also known as queen bee acid [38]. Melittin, a major component from the venom of *A. mellifera*, has also shown interesting antimicrobial activity, including in in vivo experiments with mice infected with MRSA [39].

Other animals that can live in contaminated environments and exposed to infections are reptiles, such as snakes that are able to ingest rodents infected with germs and not develop a disease. Results suggest that animals exposed to huge amounts of pathogens can be a valuable source of antimicrobial molecules. However, to further study and identification of the key molecules responsible for the activity, it is necessary to know if they would be candidates for drugs with real applicability in therapies [40]. There are studies in Black cobra (*Naja naja karachiensis*) that show that plasma lysates and certain organs have a potent antimicrobial capacity against *E. coli* K1, MRSA, *P. aeruginosa*, *Streptococcus pneumoniae*, *Acanthamoeba castellanii*, and *Fusarium solani*. Against *E. coli* K1, solutions containing 25% and 50% of plasma from the blood of the Black cobra showed a bactericidal activity of 85% and 93% respectively with respect to the effect of the antibiotic gentamicin. Against MRSA, concentrations of 25% and 50% of plasma showed activity of 90% and 93%, respectively. Lung and gallbladder lysates also showed high antimicrobial capacity against MRSA. Antimicrobial molecules can also be extracted from the venom produced by certain species of snakes, such as cathelicidines or toxins. A cathelicidin-like antimicrobial peptide (cathelicidin-BF) isolated from the venom of *Bungarus fasciatus* has shown high antimicrobial activity, including drug-resistant bacteria [41]. *Crotalus adamanteus* toxin-II (CaTx-II) exerted a strong antimicrobial effect against *S. aureus*, *Burkholderia pseudomallei* and *Enterobacter aerogenes* by causing pores and damaging their membranes. Interestingly, this compound showed no cytotoxicity against lung (MRC-5), skin fibroblast (HEPK) cells or treated mice [42].

Molecules with great antimicrobial capacity have also been found in crustaceans, coming from their immune system. The anti-lipopolysacchride factor of red claw crayfish *Cherax quadricarinatus* has shown low minimum bactericidal concentrations (MBC) against Gram-negative *Shigella flexneri* (MBC < 6 µM) and Gram-positive *S. aureus* (MBC < 12 µM), meaning a high antimicrobial capacity. Studies showed that the mechanism of action of this compound does not appear to be related to the bacterial plasma membrane alteration, requiring more studies to find its specific mechanism [43].

The venom of *Vaejovis mexicanus*, a mexican scorpion, has an AMP called vejovine, which presents a high antimicrobial capacity against MDR Gram-negative bacteria with MIC values between 4.4 µM and 50 µM [44].

#### 3.2.2. Bacterial Origin

Bacteria are the most prolific source of NPs with antimicrobial activity found so far, especially those of the actinomycetes class. Their great diversity, competitiveness and colonization capacity have led them to the development of secondary metabolites capable of giving them great advantages over other bacterial species. As described in previous sections, the detection and isolation of these bacterial antimicrobial NPs propelled medical science vertiginously in the middle of the last century. Some of the most relevant are described below.

Some of the most important antimicrobial molecules produced by bacteria of the actinomyces class are: vancomycin, baulamycin, fasamycin A and orthoformimycin. Vancomycin is a naturally occurring tricyclic glycopeptide extracted from *Streptococcus orientalis* that has reaped great success as an antibiotic against Gram-positive bacteria, especially against threats that are resistant to other treatments such as MRSA and penicillin-resistant pneumococci among others [45]. Vancomycin forms hydrogen bonds with the terminal dipeptide of the nascent peptidoglycan chain during biosynthesis of the bacterial cell wall. This union prevents the action of penicillin-binding proteins (PBPs), interrupting further wall formation and finally activating autolysin-triggered cell rupture and cell death [46]. Another important bacterial NP is produced by actinomyces is baulamycin, which is an isolated molecule of the marine bacterium *Streptomyces tempisquensis* that can inhibit the biosynthesis of iron-chelating siderophores in *S. aureus* (targeting staphylopherrin B) and *Bacillus anthracis* (targeting petrobactin), helping to treat MRSA and anthrax infections, respectively. In addition, it was also able to inhibit the growth of Gram-negative bacteria such as *S. flexneri* and *E. coli*, turning baulamycin and its derivatives into potential broad-spectrum antibiotics [47]. Fasamycin A is a polyketide isolated from *Streptomyces albus* that shows specific antimicrobial activity against Gram-positive bacteria such as vancomycin-resistant Enterococci (VRE) and MRSA with MIC values of 0.8 and 3.1 µg/mL, respectively. This molecule targets FabF in the initial condensation step of the elongation cycle from the lipidic biosynthetic bacterial metabolism [48]. Orthoformimycin is a molecule produced by *S. griseus* which can inhibit bacterial translation by more than 80% in the case of *E. coli*. Although the mechanism of action is not clear now, one hypothesis is the decoupling of mRNA and aminoacyl-tRNA in the bacterial ribosome [49].

The actinobacteria class is also prolific in the production of antimicrobial molecules. One example is kibdelomycin, which is a potent inhibitor of DNA synthesis that was isolated from *Kibdelosporangium* sp., MA7385. Its complex structure and its infrequent function as an inhibitor of bacterial DNA gyrase and IV topoisomerase make kibdelomycin the first bacterial type II topoisomerase inhibitor discovered from natural sources in more than 60 years [50]. This molecule has a broad-spectrum antimicrobial activity against aerobic bacteria, including antibiotic-resistant bacteria such as MRSA, with a MIC value of 0.25 µg/mL. In addition, this molecule has a very low resistance development rate due to its structure and way of binding with its target, at levels of other successful antibiotics such as ciprofloxacin [51]. Another example is pyridomycin, a molecule isolated from *Dactylosporangium fulvum* which has a great antimicrobial capacity against mycobacteria, a bacterium that causes tuberculosis. This disease is becoming relevant due to the appearance of bacteria resistant to the main antibiotics used for its treatment such as the InhA inhibitor isoniazid. Pyridomycin acts on the cell wall of *Mycobacterium tuberculosis* by inhibiting the production of mycolic acid by targeting NADH-dependent enoyl- (Acyl-Carrier-Protein) reductase InhA even in strains resistant to isoniazid. Pyridomycin showed minimum bactericidal concentration (MBC) values between 0.62 and 1.25 µg/mL against *M. tuberculosis* [52].

In addition to the two classes mentioned above, there are other classes of bacteria such as deltaproteobacteria, cyanophyceae or betaproteobacteria from which antimicrobial molecules have also been isolated. Myxovirecin is a macrocyclic secondary metabolite isolated from myxobacteria (deltaproteobacteria class) that possesses broad-spectrum antibacterial capacity. It seems to inhibit the production of type II signal peptidase by blocking Lpp lipoprotein processing. Myxovirecin showed very potent activity against *E. coli* DW37 with a MIC of 0.063 µg/mL [53]. Spirohexenolide A is a natural spirotetronate originally isolated from *Spirulina platensis* of the cyanophyceae class that shows antimicrobial activity against methicillin-resistant *S. aureus* by disrupting the cytoplasmic membrane, collapsing the proton motive force [54]. Teixobactin is a naturally occurring molecule produced by *Eleftheria terrae* of the betaproteobacteria class that possesses antibacterial capacity against antibiotic-resistant pathogens in infection animal models. It acts by binding to the precursors of the bacterial wall teicoic acid, causing the digestion of the cell wall by autolysins [55].

Lypoglycopeptides isolated from different bacteria show antimicrobial activity by inhibiting signal peptidase type IB (SpsB), which is a membrane-localized serine protease that cleaves the amino-terminal signal peptide from most secreted proteins. One example is actinocarbasin, a molecule isolated from *Actinoplanes ferrugineus* strain MA7383. Moreover, this molecule enhances the activity of β-lactam antibiotics against MRSA, sensitizing it to those drugs. Arylomycin is another lipoglycopeptide with bacterial type I signal peptidase inhibitory capacity which showed antibacterial activity witch MIC values in the range of 4–64 µM against Gram-positive and 8–64 µM against Gram-negative bacteria. Krisynomycin is also a lypoglycopeptide, isolated from *Streptomyces fradiae* strain MA7310, with the capacity of inhibition of SpsB [56]. 

In addition to the natural bacteria molecules with direct antimicrobial activity, there are also others capable of attacking the virulence factors caused by bacterial infections. Skyllamycins B and C are cyclic depsipeptides isolated from marine bacterial fractions with *P. aeruginosa* biofilm inhibition and dispersal activity. The ability to prevent the formation of biofilms or to disperse those already formed is of great importance since these biofilms are one of the major causes of drug resistance in nosocomial infections. These molecules do not possess a bactericidal capacity per se, but they are effective in combination with antibiotics that are not able to act in the presence of biofilms, causing them to recover their activity as in the case of azithromycin [57].

#### 3.2.3. Fungal Origin

Fungi are eukaryotic-type living things, such as mushrooms, yeasts, and molds. Currently, the existence of some 120,000 species of fungi has been accepted, however, it is estimated that the number of different species of fungi present on earth could be between 2.2 and 3.8 million [58]. This relatively unexplored kingdom is a source of antimicrobial NPs and has great potential to be studied in the future as new species are discovered and identified.

Aspergillomarasmine A is a polyaminoacid naturally produced by *Aspergillus versicolor* capable of inhibiting antibiotic resistance enzymes in Gram-negative pathogenic bacteria, such as *Enterobacteriaceae*, *Acinetobacter spp.*, *Pseudomonas spp*. and *Klebsiella pneumoniae*. This compound has been used successfully to reverse resistance in mice infected with meropenem-resistant *K. pneumoniae* thanks to the NDM-I protein, making the bacterium sensitive to the antibiotic and ending the infection [59].

Mirandamycin is a quinol of fungal origin capable of inhibiting the growth of both Gram-negative and Gram-positive bacteria, being more effective against the latter group, including antibiotic-resistant strains such as MRSA or carbapenemase-producing *K. pneumoniae*. Its mechanism of action consists in the inhibition of the bacterial metabolism of sugars, interfering with their fermentation and transport [60].

There is evidence of the antibacterial capacity of various fungal species against Gram-positive bacteria. Extracts of *Ganoderma lucidum*, *Ganoderma applanatum*, *Meripilus giganteus*, *Laetiporus sulphureus*, *Flammulina velutipes*, *Coriolus versicolor*, *Pleurotus ostreatus* and *Panus tigrinus* demonstrated antimicrobial activity in Kirby–Bauer assays against Gram-positive bacteria, such as *S. auerus* and *B. luteus* [61].

In recent times, molecules produced by various species of marine fungi have been studied, especially those that cohabit with sponges or corals. Fungal compounds with activity against antibiotic resistant bacteria have been isolated, such as lindgomycin and ascosetin, with MIC values of 5.1 µM and 3.2 µM against MRSA, respectively. These molecules were isolated from the mycelium and the *Lindgomycetae spp* culture broth from sponges found in the Baltic and Antarctic Sea [62]. Another marine fungus capable of producing antimicrobial molecules is *Pestalotiopsis* sp., isolated from the coral *Sarcophyton* sp. This fungus produces (±) -pestalachloride D, a chlorinated benzophenone derivative, which has shown antibacterial capacity against *E. coli*, *Vibrio anguillarum* and *Vibrio parahaemolyticus* with MIC values of 5, 10 and 20 μM, respectively [63]. *Trichoderma sp.* is a sponge-derived fungus from which different aminolipopeptide classes, called trichoderins, have been isolated. These molecules have a potent antimycobacterial capacity showing MIC values between 0.02 and 2.0 µg/mL against *Mycobacterium smegmatis*, *Mycobacterium bovis* BCG, and *M. tuberculosis* H37Rv in different aerobic and hypoxic conditions [64].

#### 3.2.4. Plant Origin

Plants are a great source of biomolecules with various interesting properties for humans thanks to their enormous diversity and proven safety for human health [65]. Being sessile organisms, evolution has shaped its metabolism to produce certain molecules to cope with external aggressions and infections, since they cannot flee or defend themselves [66]. The Dictionary of Natural Products lists approximately 200,000 secondary plant metabolites, of which 170,000 have unique chemical structures [67]. Some of the families of molecules with antimicrobial capacity produced by plants are alkaloids, terpenoids, and polyphenols [68].

Plants that have been used in traditional medicine in various countries of the world for thousands of years. They are currently being studied at the molecular and functional level, rediscovering their properties and explaining their mechanisms of action.

Alkaloids have been shown to possess antimicrobial capacity against various bacterial species. Although studies of the antimicrobial capacity of pure alkaloids are limited, there are several studies on the antimicrobial activity of plant extracts that contain alkaloids as their main components. Different extracts rich in alkaloids obtained from *Papaver rhoeas* have shown activity against *S. aureus*, *Staphylococcus epidermidis* and *K. pneumoniae*, the main active component being roemerine [69]. Raw alkaloid-rich extracts of *Annona squamosa* seeds and *Annona muricata* root have also shown moderate antimicrobial capacity against *E. coli* and *S. aureus* [70].

Terpenoids, along with other families of compounds, are part of plant essential oils, many of which possess antimicrobial activity. Various in vitro studies affirm that terpenoids do not possess significant antimicrobial activity per se [71]. However, they can contribute to the antimicrobial activity of complete essential oils thanks to their hydrophobic nature and a low molecular weight that allow them to disrupt the cell wall and facilitate the action of the rest of the active components [72].

Polyphenols are molecules present in plants with a function of defense against stress and have one or more phenolic groups in their chemical structure as a common feature. There is abundant literature on the antimicrobial capacity of polyphenols and extracts of plants rich in them that have bactericidal and bacteriostatic capacity against many pathogens, both Gram-positive and Gram-negative. The potential use of polyphenols as antimicrobials is widely studied to be applied in different areas such as agriculture [73], food preservation [74] and medicine [75].

There are several subfamilies within the group of polyphenols according to their differentiated chemical structures: flavonoids, hydrolyzable tannins, lignans, phenolic acids and stilbenes. In turn, the flavonoid group can be subdivided into other subfamilies: anthocyanidins, flavanones, flavones, flavonols and isoflavones [76]. Examples of flavonoids with antimicrobial activity are quercetin [77], kaempferol [78], morin [79], myricetin [80] epigallocatechin gallate [81] or galangin [82] among many others [76,83]. Other known polyphenols with good antimicrobial activity are punicalagin, which exerts both antibacterial and antibiofilm effect against *S. aureus* [80,84], and resveratrol, which has antimicrobial activity against a wide range of bacteria [75].

The growing relevance of the study of polyphenols in the clinical setting is due to their antimicrobial synergy between polyphenols and antibiotics for clinical use. Polyphenols in subinhibitory concentrations enhance the action of an antibiotic against a bacterium that was originally resistant to its effect. For example, kaempferol and quercetin, two flavonols with antimicrobial activity on their own, have also shown to increase the efficacy of the rifampicin antibiotic against rifampicin-resistant MRSA strains by 57.8% and 75.8%, respectively. The study authors blame this increase in the activity to which these polyphenols are able to inhibit the catalytic activity of topoisomerases, inhibiting DNA synthesis, with a mechanism similar to that of the ciprofloxacin antibiotic, with which they have also shown to have a synergistic activity [85]. Epicatechin gallate (ECg), a flavanol, is capable of sensitizing strains of MRSA against β-lactam antibiotics such as penicillin or oxacillin. This polyphenol can bind to the MRSA cytoplasmic membrane and cause large changes in its structure and reducing its fluidity, decoupling the functioning mechanism of the enzyme PBP2a, which is the protein responsible for resistance to β-lactam antibiotics. In addition, ECg can reduce biofilm formation and protein secretion associated with virulence factors [86]. (-)-Epigallocatechin gallate (EGCg) is another flavanol with a great capacity to enhance the effect of antibiotics that acts mainly on the cell wall directly or indirectly and on some virulence factors, such as the production of penicillinases [87]. 

Another example of synergy between polyphenols and antibiotics is the case of the combination of catechin and epicatechin gallate extracted from *Fructus crataegi* and ampicillin, ampicillin/sulbactam, cefazolin, cefepime, and imipenem/cilastatin antibiotics, which are usually ineffective against MRSA. These combinations were effective against MRSA in both in vitro and in vivo assays using mice with an established infection model. The authors stressed that the possible mechanism of action of the combination of these two polyphenols to enhance the effect of antibiotics was the accumulation of antibiotics inside the cell thanks to the inhibition of the efflux pump gene [88]. 

In addition to synergy with antibiotics, there are also studies that point to the synergy between the polyphenols themselves, such as that between EGCg and quercetin against MRSA, attributed to a co-permeabilization process that would facilitate the activity of the compounds inside of the cell [89]. Synergic activity has also been found between the polyphenols quercetin-3-glucoside, punicalagin, ellagic acid and myricetin in different proportions and combinations against *S. aureus* CECT 59 [80].

Apart from the antimicrobial use of concrete molecules of plant origin, the use of complex extracts made from different parts of plants is common and effective. Plant extracts have a great diversity in their composition, since even from the same plant multiple completely different extracts can be obtained varying the extraction conditions. Time, temperature, solvents, pressure and other parameters such as the use of ultrasound or microwave have a huge impact on the final extract composition [90]. There is numerous evidence of the antimicrobial activity of plant extracts [76,91] and the synergistic effect that exists between different phytochemicals [80] when acting against different bacteria. An example of a plant extract with potent activity against AMR bacteria are extracts from *Lantana camara* leaves against clinical isolates of MRSA, *Streptococcus pyogenes*, VRE, *Acinetobacter baumannii*, *Citrobacter freundii*, *Proteus mirabilis*, *Proteus vulgaris* and *P. aeruginosa* [92]. The ethanolic extracts of *Anthocephalus cadamba*, *Pterocarpus santalinus* and *Butea monosperma* Lam. they have also demonstrated antimicrobial activity against MDR clinical isolates of 10 different microbial species: *S. aureus*, *Acinetobacter sp.*, *C. freundii*, *Chromobacterium violeceum*, *E. coli*, *Klebsiella sp.*, *Proteus sp.*, *P. aeruginosa*, *Salmonella typhi* and *Vibrio cholerae* [93,94]. In the case of *B. monosperma* Lam., antimicrobial activity was also found in the extract made with hot water from leaf.

#### 3.2.5. Summary

As a summary, Table 1 contains all the NPs mentioned above together with their producing organism, type, target bacteria, mechanism of action, main use and references. Figure 4 shows the main molecular targets of the most relevant antimicrobial NPs.

### 3.3. Antibiotics and Plant Compounds Combinations to Get around AMR

The synergic combination of antibiotics and phytochemicals represents a promising strategy with numerous clinical and developmental benefits. Some plant compounds have direct antimicrobial activity against antibiotic-resistant bacteria, while others can sensitize resistant bacteria against antibiotics, reversing the resistance as mentioned and exemplified in the previous section. Some of these NPs can enhance the effect of antibiotics in different ways, such as facilitating their entry into the cell by destabilizing the cytoplasmic membrane [153,154], inhibiting efflux pumps (EPs) [155] or dispersing biofilms [156] among other mechanisms of action (Figure 4). Some of the synergistic interactions between phytochemicals and antibiotics include increased efficiency, lower antibiotic doses, reduced side effects, increased bioavailability and increased stability [157]. The multidimensional and multifactorial activity of phytochemicals studied by network pharmacology is crucial for synergy with clinical antibiotics, opening the door to many different potential combinations. Moreover, the use of molecules that have already passed the relevant clinical controls, as in the case of antibiotics, in combination with innocuous natural compounds facilitates the process of research and development of new potential therapies [158].

There is clear evidence of NPs capable of inhibiting efflux pumps of AMR bacteria, specifically, phytochemicals. These molecules can inhibit various efflux pumps in different pathogenic bacterial species, both Gram-positive and Gram-negative. As an example, the NorA efflux pump of *S. aureus* SA-1199-B has been effectively inhibited using baicalein plant molecules [159], capsaicin [160], indirubin [161], kaempferol rhamnoside [162] and olympicin A [163]. NorA of *S. aureus* NCTC 8325-4 was inhibited using sarothrin [164]. Cumin demonstrated antimicrobial activity on its own and also resistance modulation properties against MRSA by inhibiting LmrS efflux pump [155]. Plant molecules inhibiting the ethidium bromide efflux pump (EtBr) have also been found: 1′-S-1′-acetoxyeugenol acetate inhibits it in *Mycobacterium smegmatis* [165], catechol and catharanthine inhibits it in *P. aeruginosa* [166,167] and galotannins inhibit it in MDR uropathogenic *E. coli* [168]. The Yojl efflux pump of MDR *E. coli* has been shown to be inhibited by molecules such as 4-hydroxy—tetralone, ursolic acid and its derivatives [169] and lysergol [170]. Berberine and palmatine inhibit MexAB-OprM from clinical isolates of MDR *P. aeruginosa* [171]. There are also complete extracts of plants with EPs inhibitory activity with clear synergistic effects with antibiotics in the treatment of MDR bacterial infections. The extract made from *Rhus coriaria* seeds have shown an obvious synergistic effect with oxytetracycline, penicillin G, cephalexin, sulfadimethoxine and enrofloxacin against MDR clinical isolates of *P. aeruginosa*. This effect is mainly attributed to the inhibitory capacity of EPs of the phytochemicals present in the extract [172]. The activity of these plant molecules as inhibitors of microbial efflux pumps can act as restorers of antimicrobial susceptibility and open the door to combined antibiotic treatments, since these could exert their action more easily by not being expelled from the bacterial interior, allowing relive obsolete or discarded therapies due to this resistance mechanism [173]. A catechin, (-)-epigallocatechin gallate (EGCg), has shown sensitizing activity in *S. aureus* against tetracycline by inhibiting EPs such as Tet (K), increasing intracellular retention of the antibiotic and enhancing its effect [174]. Stilbenes also act as EPs inhibitors against antibiotic-resistant *Arcobacter butzleri*, reducing its resistance. Resveratrol and pinosylvin have also shown activity as resistance modulators being able to even reverse the resistance completely [175].

There are studies that state that certain polyphenols, such as catechins, can enter deeply into the structure of the lipid bilayer of bacterial membranes, causing significant thermotropic changes. Lipophilic hydrocarbons present in plant extracts are known to destabilize the cellular structure of the cytoplasmic membrane, increase its permeability and interact with hydrophobic portions of proteins [176]. This could explain the potentiation in the effect of certain antibiotics against resistant bacteria, as these compounds could increase antibiotic intake and interact with resistance proteins, hindering their activity. Specifically, (-)-epicatechin gallate (ECg) has a great affinity for the staphylococcal wall and its binding to it produces biophysical changes in it that are capable of dispersing the biosynthetic machinery responsible for resistance to β-lactam antibiotics [177]. This activity would explain the restoration of the sensitivity of bacteria resistant to traditional antibiotics through the use of polyphenolic compounds capable of interacting with bacterial membranes, as in the case of catechins capable of sensitizing MRSA against oxacillin and other β-lactam antibiotics thanks to its ability to integrate and interact with the cell membrane [178,179]. 

Plant extracts are also capable of exert antimicrobial activity against AMR bacteria and synergize with antibiotics. For instance, extracts of *Duabanga grandiflora* can restore MRSA’s sensitivity to ampicillin. The mechanism proposed by the researchers is that the components of this extract can decrease the expression of the mecA gene that gives rise to the resistance protein PBP2a [180]. Extracts of *Acacia nilotica*, *Syzygium aromaticum* and *Cinnamum zeylanicum* exhibited antimicrobial capacity against a panel of AMR bacteria including clinical isolates and ATCC strains. Extract of *A. nilotica* showed MIC values as low as 9.75 µg/mL against *K. pneumoniae* ATCC-700803, *Salmonella typhimurium* ATCC-13311 and *E. faecalis* ATCC-29212 [181]. Extracts of *Salvia spp*. and *Matricaria recutita* have shown great synergy with the antibiotic oxacillin [182]. The multifactorial and multi-target character of the compounds that make up plant extracts can hinder the development of resistance by bacteria [80]. The molecular promiscuity of polyphenols, their multarget activity, the possibility of obtaining complex extracts containing multiple different polyphenols, and their synergistic effect in combined use with clinical antibiotics make natural antimicrobial compounds of plant origin ideal tools to be studied from the point of view of network pharmacology in the future. The evidence found in the combination studies between plant extracts and clinical antibiotics shows a synergistic enhancement that may be key to the fight against AMR bacteria. Although the development of new synthetic antibiotics is essential to continue the fight, the sensitization of resistant bacteria by phytochemicals is also crucial to achieving effective and long-lasting therapies [158].

Infections caused by bacteria forming biofilms are extremely difficult to treat and are much less susceptible to antibiotics [183,184]. One way to enhance the effect of an antimicrobial agent is to disrupt the biofilm that certain resistant bacteria form. Studies on *P. aeruginosa* showed that many natural products can inhibit biofilm formation or disrupt the previously formed biofilm: alginate lyase [185], ursolic acid [186], zingerone [187], cranberry proanthocyanidins [188], casbane diterpene [189], manoalide [190], solenopsin A [191], catechin [192], naringenin [193], ajoene [194], rosmarinic acid [195], eugenol [196], bergamottin [197], emodin [198] and baicalein [199] among others. These natural biofilm disrupting compounds could be a very valuable tool to be incorporated into joint therapies with traditional antibiotics when treating infections caused by AMR bacteria. For example, cranberry proanthocyanidins enhanced the activity of gentamicin in an in vivo model of infection using *Galleria mellonella* [188]. In addition, some of these compounds have intrinsic antimicrobial activity on its own, which could further increase the potency of the treatment.

### 3.4. Development of Resistance to Natural Products

Historically, bacteria have managed to develop resistance to a greater or lesser extent against most antimicrobial agents used in medicine. Nevertheless, the ability of bacteria to develop a resistance mechanism against natural products is not well documented [200]. Due to the huge chemical and structural diversity among antimicrobial products of natural origin, it is often stated the difficulty for bacteria to avoid the action of NPs [201,202]. However, there are some recent studies that suggest that bacteria can develop certain levels of resistance against plant compounds, especially enteric bacteria [203]. The mechanisms of resistance behind these observations remain unknown and literature on the subject is scarce.

There are multiple mechanisms by which a bacterium can get rid of the action of an antimicrobial molecule: target alterations, expulsion or modification of the antibiotic, inactivation, reduced permeability and biofilm formation among others [204]. These resistant mechanisms can be spontaneously developed (mutations) or acquired (by transduction, transfection or conjugation processes) as shown in Figure 5. Understanding the mechanism by which bacteria can circumvent the action of antibiotics and how they acquire these capabilities is crucial to developing effective and lasting therapies. 

Depending on their properties, some products are more susceptible than others to the appearance of bacteria resistant to them. Molecules that attack highly conserved targets are less conducive to the appearance of bacteria with mutations in said targets that confer resistance to the antimicrobial in question, since modifying one or more fundamental routes or targets can imply an unbearable fitness cost for the bacteria [205]. 

On the other hand, molecules against less conserved molecular targets are more likely to promote the development of resistance mechanisms against them. Modification of less evolutionarily conserved or non-essential targets is easier for bacteria to assimilate since they have greater flexibility to modify the molecular target or adapt their metabolism without paying a high fitness cost. Although the acquisition of antimicrobial resistance mechanisms is often accompanied by reduced fitness in the absence of a selective environment, this loss of adaptive efficacy can be counteracted by compensatory mutations or modifications in epistasis [206].

Thanks to the multifactorial nature of the molecular promiscuity of naturally occurring antimicrobial compounds, bacteria experience difficulties in changing several molecular targets simultaneously [80]. Multiple simultaneous molecular changes in a bacterium to overcome the action of a multifactorial antimicrobial agent would very negatively affect its metabolism, that is, it would have a high fitness cost potentially unacceptable for its development. Likewise, mutations that carry a high fitness cost are less likely to persist in bacterial populations once the selective pressure disappears [207]. This cost would be higher if the molecular targets of the antimicrobial were highly evolutionary conserved molecules or routes, since they would be more difficult to change while maintaining the metabolic efficiency necessary for survival and competition with other living beings. Furthermore, there are studies that affirm that many of the natural antimicrobial compounds attack macromolecular structures such as the membrane or the bacterial wall and that this fact could hinder the appearance of resistance, given that they are very difficult targets to vary as a whole [208,209].

Despite the multiple possible mechanisms for acquiring existing resistances, the use of new technologies in NPs can help prevent their development. Based on new laboratory bacterial culture techniques, it has been possible to identify and isolate interesting natural compounds such as teixobactin. This molecule displays a mechanism of action that is capable of using the bacteria’s own machinery to kill itself, in a similar way to how vancomycin, a really successful antibiotic, works. No resistant mutants have been found against teixobactin. Theoretically, the generation of resistant mutants to this compound is difficult, since its target is very conserved among the eubacteria, in addition to being exposed in the outermost part of Gram-positive bacteria. In addition, as teixobactin is produced by a Gram-negative bacterium, the molecule cannot re-enter the cell and exert its action due to the presence of the outer envelope characteristic of Gram-negative bacteria. This fact is crucial in the process of the eventual development of resistance, since the producing microorganism does not use a different metabolic route to avoid the action of the antibiotic it produces. Thus, in the absence of an intrinsic resistance mechanism in the producer, horizontal transfer of resistance genes to other species susceptible to teixobactin cannot occur [55].

Vancomycin, discovered in 1958, enjoyed a period of 30 years in which no bacterium resistant to its antibiotic action was identified, thanks to its potent and unusual mechanism of action. However, during the last 20 years, *S. aureus* strains resistant to this antibiotic have been detected [210]. One of the resistance mechanisms identified is the incorporation of D-Ala-D-lactate instead of the usual D-Ala-D-Ala at the dipeptide termini of nascent peptidoglycan, considerably reducing its binding affinity and formation disruption capacity of the bacterial wall. Other resistant strains identified have a thicker cell wall with free D-Ala-D-Ala ends that can sequester vancomycin and removing it from the place where the biosynthesis of the wall occurs [211]. Despite the emergence of these and other resistance mechanisms, researchers are currently working on vancomycin derivatives that have promising qualities that allow them to circumvent these resistance mechanisms and exert their antibiotic action. An example of this is the discovery of a new vancomycin resistance mechanism mediated by the activity of Atl amidase. This inhibition produces cellular morphological changes that reduce the action of vancomycin on the main target in the biosynthesis of the wall, increasing the tolerance of the pathogen against the antibiotic without any changes at the genetic level. The discovery of this target opens the door to the design of derivatives of vancomycin with a reduced affinity for Atl, resulting in greater efficacy against MRSA [46]. Another resistance mechanism found in *S. aureus* against vancomycin is based on the thickening of the bacterial wall, which slows the penetration of vancomycin into the bacteria [212].

A possible strategy to prevent or slow the appearance of antimicrobial-resistant bacteria is the combined use of various agents that act against different molecular targets. In this way, the bacteria will have to adopt different resistance mechanisms, which would imply a greater and less likely adaptive cost. This hypothesis could support the use of plant extracts and essential oils in traditional medicine used for millennia, since these may be composed of dozens of different phytochemicals with different mechanisms of action. The combined activity of these molecules would hinder bacterial adaptation and extend the therapeutic shelf life of antimicrobial plant extracts.

Although the idea of the difficulty of acquiring resistance against complex plant extracts is widespread, some studies go in the opposite direction. It has been observed that certain antimicrobial extracts used against enterobacteria isolated from geckos from various environments in India have reduced effectiveness. The authors attribute this resistance to the variability and changing environment that has shaped the isolates collected and used in the assay. They suggest that exposure of geckos to medicinal plants may have caused a process of selecting the bacteria present in them, resulting in strains more resistant to plant compounds [203]. Mechanisms of possible resistance are not mentioned.

### 3.5. New Methodologies to Find Antimicrobial Compounds against AMR Bacteria

Currently, there are many methodologies capable of having a very positive impact on the discovery of new natural molecules with antimicrobial capacity against AMR bacteria. Some of these methodologies are the use of -omics technologies, network pharmacology, synergy studies and in silico trials.

Thanks to the -omics technologies, today it is known that genomes of bacteria such as actinomycetes are much more complex than previously thought in the mid-twentieth century and that there are multiple secondary metabolite gene clusters (SMGCs) that could produce new NPs. It is estimated that under the conditions of the classic fermentation studies for NP isolation, less than 10% of the SMGCs are active, which could be activated using genetic techniques and varying the culture conditions to reveal potential new NPs hidden inside of the “biosynthetic dark matter” [213]. By combining the progressive lowering of the massive sequencing of bacterial genomes and the advancement of the analysis and prediction software it will be possible to identify new SMGCs and their products [214,215]. The discovery and deepening of knowledge of NP-producing modular macroenzymes such as non-ribosomal peptide synthetases and polyketyde synthetases open the door to new NPs production strategies based on combinatorial biosynthesis [15]. Scientists now have greater access to soil samples and other potential sources of NPs, which significantly increases the likelihood of finding new compounds. The use of non-laboratory-dependent metagenomic techniques and the heterologous expression of DNA extracted directly from complex samples will allow the identification and production of new NPs hitherto unknown or impossible to produce [216]. 

Other new technologies such as molecular docking or virtual simulations open the door to the effective discovery of new natural antimicrobial compounds unknown so far using computers [76,217]. In silico assays allow hundreds of thousands of molecules to be screened to efficiently select leaders, greatly reducing the cost of new drug development processes. Prediction via molecular docking or virtual simulation makes it possible to predict the interactions of a molecule with its target, obtaining huge amounts of valuable information and allowing the screening of drug libraries in a short time if the necessary computing capacity is available [218,219].

Emerging studies based on network pharmacology that expand the classic single-ligand-target viewpoint provide excellent opportunities for the development of new antimicrobial compounds. The study of the network pharmacology of phytochemicals based on their molecular promiscuity and multi-target capacity can help to better understand their antimicrobial mechanisms of action and to develop more effective therapies [220]. In turn, this point also has a positive impact on synergy studies between antibiotics and phytochemicals such as those described in the previous sections, and they are currently showing such good results.

## 4. Conclusions and Future Perspectives

In conclusion, most NPs do not have sufficient therapeutic power to perform monotherapies based on them against antibiotic resistant bacteria, however, their joint application in combination therapy with traditional antibiotics could contribute to enhance their effect, reduce their dosage, side effects and improve its pharmacokinetics and pharmacodynamics properties. Natural antimicrobial products offer a promising avenue of study in the field of antibiotic development thanks to their unique properties, natural availability and enormous chemical diversity. The prospects in the discovery of new NPs with antibiotic activity are very positive. There is a tendency to revise the traditional sources of NPs that offered such good results during the “Golden Age” [221]. The use of new technologies and applications of non-existent knowledge during that age opens the door to the second era of massive discovery of molecules with remarkable and novel biological activity against AMR bacteria.

## Figures and Tables

**Figure 1 biomedicines-08-00405-f001:**
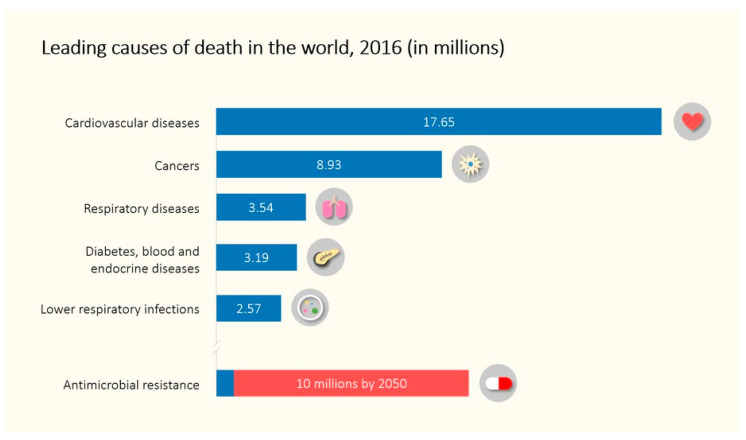
Leading causes of death in the world in 2016 (blue bars) and prognosis for antimicrobial resistance (AMR) related deaths in 2050 (red bar).

**Figure 2 biomedicines-08-00405-f002:**
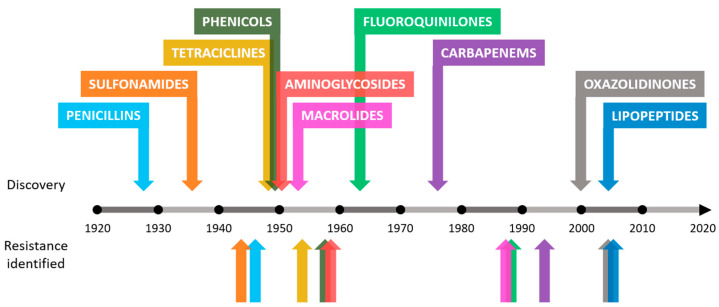
Approximate dates of discovery of new classes of antibiotics and identification of bacterial resistance.

**Figure 3 biomedicines-08-00405-f003:**
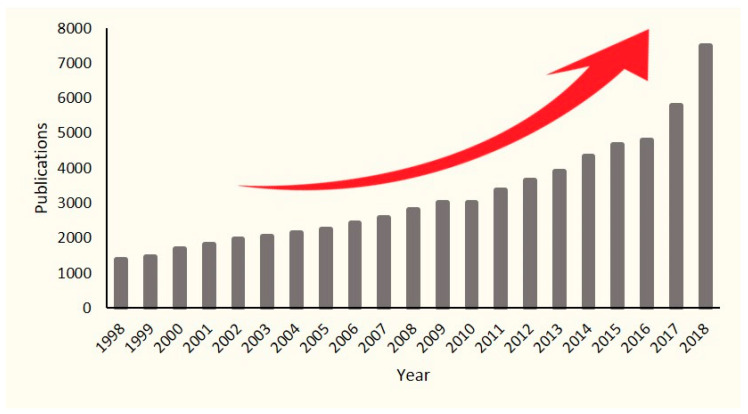
The number of research articles available in PubMed by searching “Natural Antimicrobial” from 1998 to 2018. The red arrow represents a growing trend.

**Figure 4 biomedicines-08-00405-f004:**
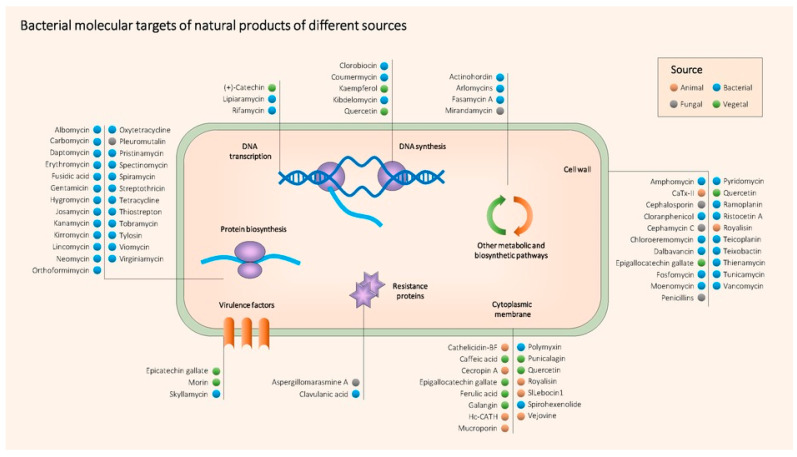
Main known molecular targets of antimicrobial NPs described in this review.

**Figure 5 biomedicines-08-00405-f005:**
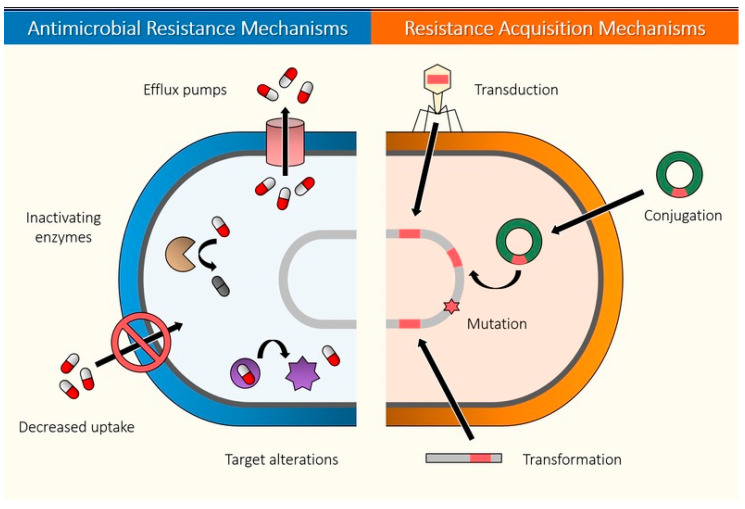
Antimicrobial resistance mechanisms and acquisition mechanisms in bacteria.

**Table 1 biomedicines-08-00405-t001:** Alphabetically ordered natural products (NPs) with their properties and capabilities. Grey shaded cells mean effectiveness against AMR bacteria. Asterisk (*) means no antimicrobial activity alone.

Natural Product	Productor Organism	Type of Organism	Activity Against	Mechanism of Action	Main Use	Reference
Actinorhodin	*Streptomyces coelicolor*	Actinomycete	Gram-positive, including multidrug-resistant *S. aureus*	ROS production inside bacterial cells	Research	[95]
Albomycin	*Streptomyces sp. ATCC 700974*	Actinomycete	Gram-negative and Gram-positive, including MRSA	Seryl t-RNA synthetase inhibition	Medicine	[96,97]
Amphomycin	*Streptomyces canus*	Actinomycete	Gram-positive, including MRSA, VRE and MDR *S. pneumoniae*	Inhibition of peptidoglycan and wall teichoic acid biosyntheses	Medicine	[98]
Apramycin	*Streptoalloteicus hindustanus*	Actinomycete	Gram-negative, including MDR *A. baumannii* and *P. areuginosa*	Inhibition of protein synthesis	Veterinary	[99]
Arlomycins	*Streptomyces* sp. Tü 6075	Actinomycete	Gram-positive and Gram-negative	Inhibition of type I bacterial signal peptidase	In research for medical use	[100]
Aspergillomarasmine A *	*A. versicolor*	Fungus	Sensitivizes carbapenem-resistant bacteria	Inhibition of bacterial metallo-β-lactamases	In research for medical use	[59]
Carbomycin	*Streptomyces halstedii*	Actinomycete	Gram-positive and *Mycoplasma*	Inhibition of protein synthesis	Medicine	[101]
Cathelicidin-BF	*Bungarus fasciatus*	Reptile	Mainly Gram-negative, including MDR strains	Damage in microbial cytoplasmic membrane	Research	[41]
CaTx-II	*C. adamanteus*	Reptile	Gram-positive and Gram-negative	Membrane pore formation and cell wall disintegration	Research	[42]
Cecropin A	*Aedes aegypti*	Insect	Gram-negative	Disruption of the cytoplasmic membrane	In research for medical use	[102]
Cephalosporin	*Cephalosporium acremonium*	Fungus	Gram-positive and Gram-negative	Inhibition of cell wall synthesis	Medicine	[103]
Cephamycin C	*Streptomyces clavuligerus*	Actinomycete	Gram-positive and Gram-negative	Inhibition of cell wall synthesis	Medicine and veterinary	[104]
Chloramphenicol	*Streptomyces venezuelae*	Actinomycete	Gram-positive and Gram-negative	Inhibition of protein synthesis	Medicine and veterinary	[105]
Chloroeremomycin	*Amycolatopsis orientalis*	Actinomycete	Gram-positive, including VRE	Inhibition of bacterial cell wall formation	Medicine	[106]
Clavulanic acid *	*S. clavuligerus*	Actinomycete	Sensitivizes β-lactam-resistant bacteria	β-lactamase inhibitor	Medicine and veterinary	[107]
Clorobiocin	*Strteptomyces roseochromogenes*	Actinomycete	Gram-positive	Inhibitors of DNA gyrase	Medicine	[108]
Coumermycin	*Streptomyces rishiriensis*	Actinomycete	Mainly Gram-positive	Inhibition of DNA gyrase	Research	[109,110]
Dalbavancin	*Nonomuraea sp.*	Actinomycete	Gram-positive, including MRSA	Inhibition of cell wall synthesis	Medicine	[111]
Daptomycin	*Streptomyces roseosporus*	Actinomycete	Gram-positive	Inhibition of protein, DNA and RNA synthesis	Medicine	[112]
Epigallocatechin gallate	Abundant in *Camellia sinensis*	Plant	Gram-positive and Gram-negative	Damage in microbial cytoplasmic membrane	In research for medical use	[81,113]
Erythromycin	*Saccharopolyspora erythraea*	Actinomycete	Gram-positive	Inhibition of protein synthesis	Medicine	[114]
Fosfomycin	*Streptomyces wedmorensis*	Actinomycete	Gram-positive and Gram-negative	Inhibition of cell wall synthesis	Medicine	[115]
Fusidic acid	*Fusidium coccineus*	Fungus	Gram-positive, including MRSA	Inhibition of protein synthesis	Medicine	[116]
Gentamicin	*Micromonospora purpurea*	Actinomycete	Gram-negative	Inhibition of protein synthesis	Medicine	[117]
Gramicidin S	*B. subtilis*	Bacillales	Gram-positive and Gram-negative	Delocalizes peripheral membrane proteins involved in cell division and cell envelope synthesis	Medicine	[118]
Hc-CATH	*Hydrophis cyanocinctus*	Reptile	Gram-positive and Gram-negative	Damage in microbial cytoplasmic membrane	Research	[119]
Hygromycin	*Streptomyces hygroscopicus*	Actinomycete	Gram-positive	Inhibition of protein synthesis	Veterinary and research	[120]
Josamycin	*Streptomyces narbonensis*	Actinomycete	Gram-positive, certain Gram-negative and *mycoplasma*	Inhibition of protein synthesis	Medicine	[121]
Kanamycin	*Streptomyces kanamyceticus*	Actinomycete	Mainly Gram-negative and certain Gram-positive	Inhibition of protein synthesis	Medicine	[122]
Kirromycin	*Streptomyces collinus*	Actinomycete	Anaerobes, *neisseriae* and *streptococci*	Inhibition of protein synthesis	Research	[123,124]
Lincomycin	*Streptomyces lincolnensis*	Actinomycete	Gram-positive	Inhibition of protein synthesis	Medicine	[125]
Lipiaramycin	*Dactosporangium aurantiacum*	Actinomycete	Gram-positive and *Mycobacterium*, including MDR strains	Inhibition of early transcription	Medicine	[126]
Melittin	*A. mellifera*	Insect	Gram-positive and Gram-negative, including MDR strains	Damage in microbial cytoplasmic membrane	Medicine	[39]
Mirandamycin	Endophytic fungus isolated from the twig of *Neomirandea angularis*	Fungus	Gram-negative and Gram-positive, including MRSA	Inhibition of bacterial quinol oxidase/ROS production	In research for medical use	[60]
Moenomycin	*Streptomyces ghanaensis*	Actinomycete	Gram-positive	Inhibition of cell wall synthesis	Veterinary	[127]
Morin	*Moraceae* family	Plant	Gram-positive and Gram-negative	Inhibition of adhesion to host tissue and DNA helicase	Food technology	[79]
Mucroporin	*Lychas mucronatus*	Arachnid	Gram-positive and Gram-negative, including MDR strains	Damage in microbial cytoplasmic membrane	Research	[128]
Neomycin	*S. fradiae*	Actinomycete	Gram-positive and Gram-negative	Inhibition of ribonuclease P	Medicine	[129]
Orthoformimycin	*S. griseus*	Actinomycete	Gram-positive and Gram-negative	Inhibition of protein synthesis	In research for medical use	[49]
Oxytetracycline	*Streptomyces rimosus*	Actinomycete	Gram-positive and Gram-negative	Inhibition of protein synthesis	Aquaculture	[130]
Penicillins	*Penicillium crysogenum*	Fungus	Gram-positive and Gram-negative	Inhibition of cell wall synthesis and activation of the endogenous autolytic system	Medicine	[131]
Pleuromutalin	*Clitopilus scyphoides*	Fungus	Gram-positive, Gram-negative and *Mycoplasma*	Inhibition of translation	Veterinary	[132]
Polymyxin	*Paenibacillus polymyxa*	Bacillales	Mainly Gram-negative (including MDR) and certain Gram-positive	Disruption of the cytoplasmic membrane	Medicine	[133]
Pristinamycin	*Streptomyces pristinaespiralis*	Actinomycete	Gram-positive, including MRSA	Inhibition of protein synthesis	Medicine	[134]
Punicalagin	Abundant in *Punica granatum*	Plant	Gram-positive and Gram-negative	Damage in microbial cytoplasmic membrane	Food technology	[80,84]
Quercetin	Ubiquitous in plants	Plant	Gram-positive and Gram-negative	Damage in the structure of the bacterial cell wall and cell membrane	In research for medical use	[135]
Ramoplanin	*Actinoplanes sp.* ATCC 33076	Actinomycete	Gram-positive, including MDR strains	Inhibition of cell wall synthesis	Medicine	[136]
Resveratrol	Abundant in grapes, berries and legumes	Plant	Gram-positive and Gram-negative, including MDR strains	Inhibition of motility, adhesion, quorum sensing, biofilm formation, flagellar gene expression and hemolytic activity	Medicine	[75]
Rifamycin	*Amycolatopsis mediterranei*	Actinomycete	Gram-positive and certain Gram-negative	Inhibition of DNA-dependent RNA synthesis	Medicine	[137]
Ristocetin A	*A. orientalis*	Actinomycete	Gram-positive, including MRSA	Inhibition of cell wall synthesis	Medicine	[138]
Royalisin	*Apis melifera*	Insect	Mainly gram-positive	Damage in the structure of the bacterial cell wall and cell membrane	Research	[37]
Skyllamycins	*Streptomyces sp. KY 11784*	Actinomycete	Gram-positive	Inhibition of biofilm formation	In research for medical use	[139]
SlLebocin1	*Spodoptera litura*	Insect	Gram-positive and Gram-negative	Damage in microbial cytoplasmic membrane or cell division inhibition	Research	[140]
Spectinomycin	*Streptomyces spectabilis*	Actinomycete	Gram-positive and Gram-negative	Inhibition of protein synthesis	Medicine	[141]
Spiramycin	*Streptomyces ambofaciens*	Actinomycete	Gram-positive and Gram-negative	Inhibition of protein synthesis	Medicine	[142]
Streptothricin	*Streptomyces (multiple species)*	Actinomycete	Gram-positive and Gram-negative	Inhibition of protein synthesis	Veterinary and plant production	[143]
Teicoplanin	*Actinoplanes teichomyceticus*	Actinomycete	Gram-positive, including MRSA	Inhibition of bacterial cell wall synthesis	Medicine	[144]
Teixobactin	*Eleftheria terrae*	Betaproteobacteria	Gram-positive, including MRSA	Causes digestion of the cell wall by autolysins	Medicine	[55]
Tetracycline	*Streptomyces rimosus*	Actinomycete	Gram-positive and Gram-negative	Inhibition of protein synthesis	Medicine	[145]
Thienamycin	*Streptomyces cattleya*	Actinomycete	Gram-positive and Gram-negative	Inhibition of bacterial cell wall synthesis	Derivates used in medicine	[146]
Thiostrepton	*Streptomyces azureus*	Actinomycete	Gram-positive and Gram-negative	Inhibition of protein synthesis	Veterinary and research	[147]
Tobramycin	*Streptoalloteicus hindustanus*	Actinomycete	Gram-negative	Inhibition of protein synthesis and membrane destabilization	Medicine	[148]
Tunicamycin	*Streptomyces chartreusis*	Actinomycete	Gram-positive	Inhibition of peptidoglycan and lipopolysaccharide synthesis	Research	[149]
Tylosin	*S. fradiae*	Actinomycete	Gram-positive and *Mycoplasma*	Inhibition of protein synthesis	Veterinary	[150]
Vancomycin	*S. orientalis*	Actinomycete	Gram-positive, including MRSA	Inhibition of bacterial cell wall synthesis	Medicine	[45]
Vejovine	*V. mexicanus*	Arachnid	Gram-negative, including MDR	Damage in microbial cytoplasmic membrane	Research	[44]
Viomycin	*Streptomyces sp. 11861*	Actinomycete	MDR *Mycobacterium*	Inhibition of protein synthesis	Medicine	[151]
Virginiamycin	*Streptomyces virginiae*	Actinomycete	Gram-positive	Inhibition of protein synthesis	Agriculture and industry	[152]

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
