# Peer review of "Tackling Antibiotic Resistance with Compounds of Natural Origin: A Comprehensive Review"

_biomedicines, 2020, doi:10.3390/biomedicines8100405_

Round 1
Reviewer 1 Report
The article is based on the impressive bibliography searching, it is also well written and comprehensive, despite its lenght.
However, I suggest addition definition/ definitions of "antimicrobial activity". In cases of refering to some studies Authors give information about MIC values for given substances, but not for all. What I mean is to give information what method are/ were used for assessing antimicrobial activities of given NPs, if these methods have a kind of standarization or are settled by the researchers.
For example for testing susceptibility of the bacterial strains for specific antibiotics CLS or EUCAT method can be used as a referential and repetitive. In my opinion giving short information about the method of antimicrobial properties of NPs testing would be valuable.
Additionally, please check if "gram-positive" and "gram-negative" is correctly writtne. In my opinion it should be "Gram".
Author Response
Response to Reviewer 1 Comments
Point 1: I suggest addition definition/ definitions of "antimicrobial activity". In cases of refering to some studies Authors give information about MIC values for given substances, but not for all. What I mean is to give information what method are/ were used for assessing antimicrobial activities of given NPs, if these methods have a kind of standarization or are settled by the researchers.
For example for testing susceptibility of the bacterial strains for specific antibiotics CLS or EUCAT method can be used as a referential and repetitive. In my opinion giving short information about the method of antimicrobial properties of NPs testing would be valuable.
Response 1: First, we greatly appreciate the comments the reviewer and his/her efforts to improve our manuscript.
In order to clarify aspects related to definitions and susceptibility tests normalization, we added the following paragraph in the Methodology section (line 88), as suggested by the reviewer:
The term "antimicrobial activity" is used throughout this work to refer to the process of killing or inhibiting the growth of microbes. Usually this activity is expressed as MIC (minimum inhibitory concentration) values for a given agent. The methods to test microbial susceptibility compiled in this work are in accordance with the guidelines of the European Committee on Antimicrobial Susceptibility Testing (EUCAST) and The Clinical & Laboratory Standards Institute (CLSI). Following the EUCAST guidelines for the reproducibility and reliability of antimicrobial assays, broth dilution or microdilution methods should be used to test microbial susceptibility [14].
Point 2: please check if "gram-positive" and "gram-negative" is correctly writtne. In my opinion it should be "Gram".
Response 2: Following the CDC directions (https://wwwnc.cdc.gov/eid/page/preferred-usage) on the use of the term Gram:
"Gram should be capitalized and never hyphenated when used as Gram stain; gram negative and gram positive should be lowercase and only hyphenated when used as a unit modifier. Examples:
Gram staining
gram negative
gram-positive bacteria"
We have revised the manuscript and adjusted the text to these guidelines.
Reviewer 2 Report
The present manuscript summarize the antimicrobial activity of the most
representative natural products of animal, bacterial, fungal and plant origin. Their mechanism of actions and possible resistance development are discussed.
The manuscript is well-organized, clear and it is fully consistent with the journal’s aims and scope The topic is very actual due to the still increasing level of antibacterial resistance. I recommend to publish this review in current form.
Author Response
Response 1: It makes us very proud to read this. Thank you very much.